# RNA-Seq Analysis of Extradomain A and Extradomain B Fibronectin as Extracellular Matrix Markers for Cancer

**DOI:** 10.3390/cells12050685

**Published:** 2023-02-21

**Authors:** Ryan C. Hall, Amita M. Vaidya, William P. Schiemann, Quintin Pan, Zheng-Rong Lu

**Affiliations:** 1Department of Biomedical Engineering, Case Western Reserve University, Cleveland, OH 44106, USA; 2Case Comprehensive Cancer Center, Case Western Reserve University, Cleveland, OH 44106, USA; 3University Hospitals Seidman Cancer Center, Cleveland, OH 44106, USA; 4Department of Otolaryngology-Head and Neck Surgery, School of Medicine, University Hospitals, Case Western Reserve University, Cleveland, OH 44106, USA

**Keywords:** extracellular matrix, tumor microenvironment, fibronectin, extradomain A fibronectin, extradomain B fibronectin

## Abstract

Alternatively spliced forms of fibronectin, called oncofetal fibronectin, are aberrantly expressed in cancer, with little to no expression in normal tissue, making them attractive biomarkers to exploit for tumor-targeted therapeutics and diagnostics. While prior studies have explored oncofetal fibronectin expression in limited cancer types and limited sample sizes, no studies have performed a large-scale pan-cancer analysis in the context of clinical diagnostics and prognostics to posit the utility of these biomarkers across multiple cancer types. In this study, RNA-Seq data sourced from the UCSC Toil Recompute project were extracted and analyzed to determine the correlation between the expression of oncofetal fibronectin, including extradomain A and extradomain B fibronectin, and patient diagnosis and prognosis. We determined that oncofetal fibronectin is significantly overexpressed in most cancer types relative to corresponding normal tissues. In addition, strong correlations exist between increasing oncofetal fibronectin expression levels and tumor stage, lymph node activity, and histological grade at the time of diagnosis. Furthermore, oncofetal fibronectin expression is shown to be significantly associated with overall patient survival within a 10-year window. Thus, the results presented in this study suggest oncofetal fibronectin as a commonly upregulated biomarker in cancer with the potential to be used for tumor-selective diagnosis and treatment applications.

## 1. Introduction

Cancer kills upwards of 600,000 people in the United States every year, making it the second leading cause of death in the United States [1]. Early detection and cancer-specific treatment have great potential to significantly improve the survival of cancer patients and reduce cancer-related mortality [2]. Cancer cells are biologically heterogeneous and dynamic in nature, presenting tremendous challenges in the development of molecular imaging technologies and therapies that target cellular markers [3]. The tumor microenvironment (TME) plays a crucial role in cancer development, progression, and treatment sensitivity [4,5,6]. Many cancer types share similar TME features, including angiogenic tumor vasculature, connective tissues, immune microenvironment, and extracellular matrix (ECM) [7,8,9,10]. Oncogenic markers in the TME are attractive targets for the design and development of molecular imaging technologies and targeted cancer therapeutics for early cancer detection and imaging-guided precision healthcare for cancer patients [11,12].

Compared to normal tissue, the tumor ECM is often highly enriched with aberrantly expressed proteins [13]. While most ECM proteins are expressed in normal tissues and tumors, some undergo tissue-specific alternative splicing, where specific exons or mRNA fragments can be included or excluded from the final mRNA transcript and translated protein [14]. Fibronectin (FN) is an ECM glycoprotein essential to normal tissue biology and is known to undergo alternative splicing [15]. Two alternatively spliced FN exons—extradomain-A (EDA) and extradomain-B (EDB)—are involved in developmental and remodeling processes, such as embryogenesis, wound healing, and neovascularization [16,17,18]. Previous studies focusing on EDA- and EDB-containing FN (EDA-FN and EDB-FN, respectively) in human tumor specimens have found them to be aberrantly expressed in several cancers, including breast cancer, head and neck cancer, pancreatic cancer, and prostate cancer, among others [19,20,21,22,23,24,25]. Furthermore, high expression of EDA-FN and EDB-FN in tumors has been associated with angiogenesis, epithelial-to-mesenchymal transition (EMT), tumor cell migration and invasion, and therapy resistance [26,27,28,29]. On account of their significant roles in fetal development and tumor-related processes, EDA-FN and EDB-FN are called “oncofetal” FNs. Preclinical work in several cancer types has already exploited the aberrant expression of oncofetal FN in tumors for targeted diagnostic and therapeutic purposes with high tumor selectivity and efficacy [30,31,32]. However, while these studies and clinical evidence have explored specific cancer types based on limited sample sizes, there has yet to be a large-scale pan-cancer study focusing on alternative splicing characteristics of FN in the context of clinical diagnostics and prognostics to examine its potential as a targetable biomarker for molecular diagnostics and therapies.

To assess FN expression and its oncofetal subtypes from a pan-cancer perspective, large, publicly available datasets can be explored. The Cancer Genome Atlas (TCGA) and the Genotype-Tissue Expression (GTEx) projects have collected thousands of cancer and normal tissue samples for a variety of tissue sites for large-scale, big-data analysis (Table 1, Appendix A) [33,34]. These projects conducted RNA-Seq analysis on mRNA samples from the collected tissues, producing large arrays of data mapped to the Ensembl genome library [35]. To utilize the projects in a single dataset, the UCSC TOIL Recompute project reprocessed the RNA-Seq samples to ensure consistent meta-analysis between datasets [36,37]. Using the TOIL Recompute results, gene and transcript expression can be explored in the context of clinical information provided with the samples to analyze differences in alternative splicing characteristics between normal tissue samples and tumors. Thus, this study investigates the aberrant expression of FN and its alternative splicing characteristics in primary tumors and normal tissues, with emphasis on its utility as a targetable biomarker.

## 2. Results

### 2.1. Fibronectin Exhibits Aberrant Expression in Primary Tumor Tissue

FN expression was first explored at both the mRNA (*FN1* gene) and protein levels to elucidate correlations between transcription and translation activity. In the TCGA database, FN is one of the only ECM proteins for which mRNA and protein expression data are both provided for primary tumors. FN mRNA expression in primary tumors follows a significant positive relationship with FN protein expression (Figure 1A, Appendix A). This suggests that FN mRNA expression can be used as an approximate analog of downstream FN protein expression. Since protein expression data are not provided for normal tissue samples or specific FN isoforms, and given the positive correlation between FN mRNA and protein expression, further analyses in this study will focus on FN mRNA expression.

FN is known to generally exhibit upregulated expression in tumor tissues compared with corresponding normal tissues. The GTEx library provides thousands of normal tissue samples that can be added to those provided by TCGA for a more rigorous comparison of normal tissues and primary tumors. The assignments of GTEx tissue sites to corresponding TCGA tissue sites are shown in Table 1, and sample sizes for all cohorts used in this study are detailed in Appendix A. Based on data provided by TCGA and GTEx, FN mRNA exhibits higher expression in primary tumors than in normal tissues in 17 out of 25 cancer types (68%) for 10 or more normal tissue and primary tumor samples (Figure 1B, Appendix A). On average, FN mRNA exhibits 5.23× overexpression in primary tumors compared to normal tissues among the cancer types analyzed, with the highest overexpression occurring in breast cancer (BRCA, 8.4×), glioblastoma multiforme (GBM, 18.5×), head and neck squamous cell carcinoma (HNSC, 7.4×), pancreatic adenocarcinoma (PAAD, 20.9×), and thyroid cancer (THCA, 41.2×) (Figure 1C).

However, despite the differences in expression of FN mRNA between normal tissues and primary tumors, there remain issues with FN as a potential oncotarget in general. First and foremost, as a primary component of the ECM, FN also exhibits high expression in normal tissues, raising concerns about off-target effects. Second, while upregulated FN is associated with primary tumors and tumor progression, aberrant FN expression can be related to many other physiological responses and conditions, such as transitory and chronic inflammation, fibrosis, and tissue repair. Therefore, a target more selective to tumors than general FN is ideal.

### 2.2. The Fibronectin Alternative Splicing Landscape

The FN gene contains 47 exons, 3 of which exhibit alternative splicing (Figure 2A). These three exons are EDA and EDB, which are either absent or present, and the variable region (IIICS), which can take on five isoforms of differing amino acid lengths (V0, V64, V89, V95, and V120) (Figure 2B). The EDA and EDB domains are of particular interest due to their association with oncogenic processes. In total, all potential combinations of alternative splicing (EDA−/+, EDB−/+, and V0/64/89/95/120) result in a maximum of 20 full-length FN proteins (Appendix A). The most recent mRNA mapping to the human genome conducted by Ensembl identified 27 FN transcripts, 10 of which correspond to full-length FN, called ECM fibronectin (ECM-FN) for the remainder of this study (Figure 2D). Of the 10 ECM-FNs, 5 contain EDA and 3 contain EDB. For analyses conducted in this study, the combined expressions of the 10 ECM-FNs, 5 full-length EDA-containing transcripts (EDA-FNs), and 3 full length EDB-containing transcripts (EDB-FNs) are examined in more detail (Figure 2C).

ECM-FN accounts for approximately 21% of total FN mRNA in normal tissues, representing a significantly small proportion of total FN mRNA expression (Figure 3A). In addition, ECM-FN follows similar overexpression patterns in primary tumors to the *FN1* gene as a whole. Of the 25 cancer types analyzed, 19 (76%) overexpress ECM-FN in primary tumors compared to normal tissue (Figure 3B, Appendix A). Furthermore, primary tumors exhibit 6.60× overexpression of ECM-FN relative to normal tissue, which is substantially higher than the 5.23× overexpression of the *FN1* gene (Figure 3C).

Of the remaining 17 FN transcripts, just 1 corresponds to a known truncated protein called migration stimulating factor (MSF), which is a soluble protein present in blood serum and the stroma (Appendix A). The rest are likely to represent truncated transcripts or misaligned reads. While these short transcripts do not have a corresponding full-length protein, three contain EDA and two contain EDB. For analyses conducted in this study, the expression of total FN mRNA (i.e., all transcripts/*FN1* gene), all EDA-containing transcripts (Exon A; five full-length and three short) and all EDB-containing transcripts (Exon B; three full-length and two short) are also examined (Figure 2C).

### 2.3. Oncofetal Fibronectin Exhibits Aberrant Expression in Primary Tumor Tissue

The expression of EDA and EDB in FN was next explored in normal tissue and primary tumors. When analyzing all available FN transcripts, both Exon A and Exon B represent significantly small proportions of the *FN1* gene as a whole (Figure 4A). In primary tumors, whereas the *FN1* gene exhibited an average overexpression of 5.23× compared to normal tissue, Exon A and Exon B both exhibited significantly higher overexpression at 6.77× and 6.03×, respectively (Figure 4B, Appendix A). When looking at individual cancer types, Exon A exhibited higher normalized expression than the *FN1* gene in 17 out of 25 (68%) cancer types, while Exon B exhibited higher normalized expression in 19 out of 25 (76%) cancer types (Figure 4C).

When narrowing our analyses strictly to ECM-FNs, both EDA-FN and EDB-FN likewise represent significantly small proportions of ECM-FNs as a whole (Figure 4D). In primary tumors, whereas ECM-FN exhibits an average overexpression of 6.60× compared to normal tissues, EDA-FN and EDB-FN exhibited substantially higher overexpression at 7.45× and 9.28×, respectively (Figure 4E, Appendix A). Examining individual cancer types revealed that EDA-FN exhibits higher normalized expression than ECM-FNs in 15 out of 25 (60%) cancer types and that EDB-FN likewise exhibits higher normalized expression in 16 out of 25 (64%) cancer types (Figure 4F).

Under both analysis schemes, FN transcripts containing EDA and EDB exhibited a greater level of overexpression in primary tumors compared to the *FN1* gene (in the cases of Exon A and Exon B) and ECM-FN (in the cases of EDA-FN and EDB-FN). This indicates that oncofetal FNs could serve as potential tumor-selective ECM targets. The remainder of this study will focus on the correlations between oncofetal FN expression and clinical diagnostic and prognostic factors. As the focus will be on FN expressed in the ECM, only ECM-FNs and the corresponding EDA-FNs and EDB-FNs containing transcripts will be investigated in more detail.

### 2.4. Oncofetal Fibronectin Expression and Early-Stage Cancer

One of the leading prognostic indicators for many cancer types is tumor stage at the time of diagnosis. Several cancers already have clinically viable screening mechanisms to detect in situ or early-stage disease, but many others do not. TCGA provides pathologic staging information for most of the samples in the database. After grouping patients by stage, it is readily visible that several cancer types in TCGA are diagnosed with stage I disease less than 20% of the time (Figure 5A). While breast cancer patients in this analysis exhibited low rates of early-stage diagnosis, the clinical rate of early-stage diagnosis for breast cancer is approximately 48% [38].

Looking strictly at the cancer types with low early-diagnosis rates, patient survival is significantly better when tumors are diagnosed at stage I than at all other stages (Figure 5B). In these cancer types, both EDA-FN and EDB-FN expression were borderline significantly higher on average in stage I tumors compared with corresponding normal tissues (Figure 5C). Pancreatic cancer, with the highest overexpression of oncofetal FN in stage I tumors, has no clinically viable screening procedures. As such, overexpression of EDA-FN and EDB-FN in early-stage pancreatic tumors presents an opportunity for targeting the ECM to provide better mechanisms and procedures for identifying tumors at an early stage. Similarly, although screening procedures exist for breast cancer and head and neck squamous cell carcinoma, the high overexpression of oncofetal FN in stage I tumors presents a potential ECM marker to exploit for supplementing current clinical screening procedures and improving early detection of disease.

### 2.5. Oncofetal Fibronectin Expression and Lymph Node Activity

Lymph node status is another factor highly predictive of patient prognosis, which is especially true in head and neck squamous cell carcinoma (HNSC), where it represents the most significant prognostic indicator [39]. In the TCGA database, patients can be grouped into those diagnosed with negative lymph node activity (N0) and those diagnosed with positive activity (N+). HNSC patients diagnosed N+ exhibited significantly poorer overall survival than patients diagnosed N0, with median survival times of 2.99 and 7.41 years, respectively (Figure 6A). As such, ensuring that the clinical diagnostic processes for lymph node activity are as accurate and sensitive as possible is of the utmost importance for HNSC patients.

Oncofetal FN expression was next compared between normal tissue samples and primary tumors based on pathologic N stage. Samples from patients diagnosed N0 exhibited significantly higher EDA-FN and EDB-FN expression than normal tissue samples. Furthermore, patients diagnosed N+ exhibited significantly higher EDA-FN and EDB-FN expression than N0 patients, thereby representing a stepped trend of increasing oncofetal FN expression with increasing pathologic N stage (Figure 6B). In addition, when expanding the analysis of pathologic lymph node activity to other cancers, N+ patients exhibit significantly higher EDA-FN and EDB-FN expression than N0 patients on average, suggesting the potential for oncofetal FNs as unique molecular targets for aiding in the predictive diagnosis of lymph node activity (Appendix A).

To further examine this, a binary classification test was performed. HNSC is the only cancer type in the TCGA database for which clinical and pathologic N stages are both supplied for a majority of samples in the dataset. In this test, pathologic N stage (pN), which is determined via lymph node dissection and subsequent histological examination, represents the ground truth, while clinical N stage (cN), which is determined via physical and radiological examination, is the clinical test. The binary classification included 400 HNSC patients, with a prevalence of positive lymph node activity of 58.3% (Figure 6C, Appendix A). Clinical N staging correctly staged 139 patients as N0 and 175 patients as N+, yielding an accuracy of 78.5% (Figure 6C, Appendix A). However, 58 patients who were clinically N0 were pathologically N+, representing understaged patients, while 28 patients who were clinically N+ were pathologically N0, representing overstaged patients (Figure 6C, Appendix A). Looking strictly at patients who were incorrectly diagnosed, both EDA-FN and EDB-FN expression were borderline significantly higher in understaged patients compared to those that were overstaged (Figure 6D). This result is in agreement with the overall expression trends observed when analyzing all N0 and N+ patients shown in Figure 6B. With this in mind, targeting oncofetal FN presents on opportunity to enhance the diagnostic procedures for lymph node activity in head and neck squamous cell carcinoma and has the potential to be applied to other cancers with high rates of lymph node activity as well.

### 2.6. Oncofetal Fibronectin Expression and Histological Grade

Just like overall tumor stage, histological grade is closely correlated with patient prognosis. In all cases in which a histological grade is determined, the tumors must be biopsied, which is often a highly invasive procedure. Diagnostic methods that exploit the TME have the potential to provide additional information about tumor morphology under non-invasive conditions that could aid in the diagnostic process and help limit the number of invasive biopsies required. This is especially true in brain cancer, where tumors range from low-grade gliomas to high-grade glioblastomas. The TCGA database provides data for two brain cancers—low-grade glioma (LGG) and glioblastoma multiforme (GBM)—which can be compared to determine differences in FN expression associated with histological grade. Brain cancer prognosis strongly correlates with tumor grade at the time of diagnosis (Figure 7A). Examining the expression of oncofetal FN reveals that both EDA-FN and EDB-FN follow a stepped trend of significantly increasing expression as brain cancer grade increases (Figure 7B).

Similarly, prostate cancer patients are nearly universally biopsied and diagnosed with a Gleason score based on tumor morphology, where higher Gleason scores generally correlate with poorer prognosis. Although prostate cancer patients have a high overall survival rate, there is a notable difference in survival between low-to-mid-grade tumors and high-grade tumors, demonstrating the significance of diagnosing Gleason score to help determine patient prognosis (Figure 7C). Examining oncofetal FN expression shows that both EDA-FN and EDB-FN exhibit a trend of increasing expression as Gleason score increases (Figure 7D). While the steps between Gleason scores are not significant, the overall trend of increasing expression approaches significance. 

Other cancers in the TCGA database also include histological grading information. When expanding the scope of analysis to include these other cancers, the trend of increasing oncofetal FN expression with increasing histological grade remains consistent (Appendix A). Unsurprisingly, highly aggressive cancers, such as pancreatic cancer and head and neck squamous cell carcinoma, exhibited the largest increases in expression with tumor grade. Oncofetal FNs therefore present targetable ECM biomarkers to supplement the diagnostic process in cancers that routinely undergo highly invasive biopsies.

### 2.7. Oncofetal Fibronectin Expression and Patient Prognosis

Aside from correlations between FN expression and clinical diagnostic information, FN expression has been shown to correlate directly with patient survival in several cancer types. The TCGA database provides survival information for most of the patients included in the study, so patients can be grouped based on high (top 33%) or low (bottom 33%) FN expression within each cancer type. In this analysis, there is a clear and significant trend of reduced median survival in patients expressing high EDA-FN and EDB-FN compared to those with low expression (Figure 8A,B). On average, patients expressing high levels of EDA-FN exhibited approximately 29% reduced median survival time compared to the low-expression group. Similarly, patients expressing high levels of EDB-FN exhibited approximately 26% reduced median survival time on average. Some of the largest reductions in median survival time are found in bladder cancer, esophageal cancer, mesothelioma, stomach adenocarcinoma, lung squamous cell carcinoma, and low-grade glioma (Appendix A).

The low- and high-oncofetal-FN-expression groups for all cancer types in the TCGA database, when combined, generate a pan-cancer dataset for analyzing overall survival among all cancers. It can be clearly seen that, over at least 10 years, the high-EDA-FN- and high-EDB-FN-expression groups exhibit significantly poorer survival than the respective low-expression groups (Figure 8C,D). The median survival time of patients expressing high EDA-FN is 6.23 years, representing a 2.1-year drop from the 8.35 years for the low-expression group (Figure 8C). Similarly, the median survival time for patients expressing high EDB-FN is 6.35 years, representing a 1.4-year drop from the 7.91 years for the low EDB-FN group (Figure 8D). These data suggest that oncofetal FN expression levels correlate with patient prognosis, indicating that they could serve as potential ECM targets for risk stratification and more accurate prognosis.

## 3. Discussion

Due to the heterogeneity and dynamic nature of tumor cells, there is generally no single cellular marker for targeting tumors. As such, tumor ECM markers have become objects of increased interest as more reliable targeting platforms than tumor cells themselves. FN has become an attractive ECM marker to explore for cancer molecular imaging and targeted therapy in light of its association with tumorigenesis and tumor progression. Indeed, oncofetal FNs have been investigated as tumor-selective targets due to their low expression levels in normal tissues, facilitating tissue-selective uptake and reducing negative off-target effects. Several targeting ligands, including the L19 and F8 antibodies, the ZD2 and PL1 peptides, and the EDB aptide, have been developed to selectively target oncofetal FN [21,40,41,42]. Interestingly, most ligands targeting oncofetal FN that have received extensive study have targeted EDB-FN.

The L19 antibody and its corresponding small-chain variable fragment (scFv), specific to EDB-FN, have been used extensively for cancer imaging applications [30]. Several clinical trials have been conducted for the development of L19-conjugated iodine-based agents. The L19 antibody has also been extensively used in cancer therapy applications, including radiotherapy, immunotherapy, and chemotherapy, with several other clinical trials completed or under way [30]. Similarly, the F8 antibody, which is specific to EDA-FN, has been studied as a ligand for tumor-selective immunotherapies [30].

Peptides are also attractive ligand designs due to their small size and ease of conjugation. ZD2, a seven-amino acid peptide targeting EDB-FN, has been used extensively as a targeting ligand for imaging agents and therapeutics [21,43,44,45,46,47,48,49,50,51,52]. The ZD2-targeted MRI contrast agent, MT218, has demonstrated effectiveness at doses below clinical levels in several cancers and is undergoing clinical development [53]. Similarly, an aptide developed for EDB binding has been used for targeted chemotherapy, gene delivery, and imaging agents [42,54,55,56]. Furthermore, PL1, a 12-amino acid peptide simultaneously targeting both EDB-FN and tenascin-C, has recently been developed to deliver iron oxide nanoworms to brain cancers [41].

Molecular imaging of aberrantly expressed oncofetal FNs presents an opportunity to non-invasively measure their expression in primary tumors for early cancer detection and characterization. This is especially true for pancreatic cancer, which has no clinically viable screening techniques for early-stage disease. With a 5-year survival rate of approximately 11%, detection of pancreatic cancer at an early stage provides an overwhelming benefit in terms of patient survival [1]. Low survival rates in pancreatic cancer are due, in large part, to the inability to resect late-stage tumors, as only 10–20% of pancreatic cancers are suitable for resection—an issue that early-stage diagnosis can help ablate [57]. In this study, oncofetal FN was shown to exhibit significant upregulation in stage I pancreatic tumors, suggesting a potential avenue for targeting and localizing early-stage disease via molecular imaging.

Similarly, oncofetal FN also exhibits significantly elevated expression in head and neck squamous cell carcinoma and other cancer types with positive lymph node activity, suggesting an application for diagnosing primary tumors likely to exhibit positive lymph nodes or identifying positive lymph nodes directly. As the most significant prognostic indicator for HNSC patients, the accuracy of lymph node diagnosis plays a key role in determining the courses of treatment and the expected outcomes for patients [39]. Unfortunately, clinical methodologies for determining lymph node activity in HNSC reach an accuracy of approximately 80%, meaning that upwards of a fifth of HNSC patients are incorrectly diagnosed [58]. The results of incorrect diagnoses often include unnecessary elective neck dissections for false-positive patients and the failure to provide adequate treatment for false-negative patients [58]. Thus, ensuring that diagnostic tests are as accurate as possible is of the utmost importance, and targeting oncofetal FN has the potential to provide tumor-selective markers to enhance the accuracy of these tests.

Oncofetal FNs are known markers of EMT, which is associated with cancer invasiveness. Thus, they have the potential to serve as markers to monitor disease progression as well as provide risk assessment, even in cancers with low overall FN expression. For example, prostate cancer is the most common non-cutaneous cancer and the second most common cause of death in men [1]. It is highly heterogeneous, and early detection of high-risk prostate cancer is crucial for timely treatment. Although EDA-FN and EDB-FN expression in prostate cancer samples were similar to expression in normal tissues, clinically significant prostate cancers (Gleason score ≥ 7) show a trend of increased EDA-FN and EDB-FN expression with increasing Gleason score, indicating that they are potential markers for the risk stratification of prostate cancer.

Furthermore, the association of oncofetal FN overexpression with cancer angiogenesis, EMT, and invasion suggests FN downregulation as a mechanism to control cell behavior and improve patient survival. Specifically, knocking down the expression of oncofetal FN in tumors has the potential to limit disease progression and spread, while also playing a role in improving tumor sensitivity to therapeutics. Several studies exploring the treatment of various cancers with microRNAs from the miR-200 family have demonstrated this [51,59,60,61]. FN represents one of the primary targets of the miR-200 family, among several others that directly affect pathways related to EMT [62]. In animal models, miR-200 treatment generally reduces tumor proliferation and metastasis, while also abrogating drug resistance. While these effects are not singly tied to FN expression, as miRNAs can simultaneously target dozens of mRNA transcripts, ex vivo analyses of miR-200-treated tumors often show strong knockdown of FN [51]. The results from these studies suggest that FN plays a key role in tumor progression, spread, and treatment sensitivity, which is in agreement with the clinical prognostic results of our TCGA study.

Oncofetal FN can also be utilized as a biomarker to monitor the effectiveness of traditional chemotherapies. Upon consecutive doses of chemotherapy, drug resistance is commonly developed. The dynamic nature of tumor cells generally produces cells that are susceptible to chemotherapy at certain doses, while also producing cells that are able to resist the treatment [63]. This often results in a tumor mass that initially recedes when the sensitive cells die, after which the resistant cells proliferate [64]. In many cases where drug resistance develops, ECM proteins such as oncofetal FN are known to become substantially more upregulated than the pretreatment tumor, thereby producing a more aggressive and difficult-to-treat primary tumor [65]. The enhanced oncofetal FN expression after the development of resistance presents an opportunity to use ECM biomarkers to monitor the efficacy of the therapy. In this study, the RNA-Seq analysis of oncofetal FN expression grouped all primary tumors together, as too few post-treatment samples existed for robust comparisons with pre-treatment expression. Further studies can focus on assessing the expression of tumor ECM proteins at various stages of treatment and with various responses to treatment to establish oncofetal FNs as markers for monitoring tumor progression and response to therapies.

This analytical study has multiple limitations. First and foremost, while the gene and protein expression of FN exhibit overall positive trends, there is no guarantee of direct correlation in all of the samples analyzed. Many factors contribute to protein translation from mature mRNA, which can result in large differences between mRNA and protein expression. Second, the TCGA database only provides protein expression for all collective FN proteins as a whole, so the mRNA expression of alternatively spliced transcripts cannot be verified by protein expression from the database. It is assumed, due to the positive correlation between FN gene and protein expression, that a similar trend would exist for the alternative splice variants as well. Finally, our analysis of the diagnostic and prognostic importance of FN focused strictly on ECM-FNs. While these transcripts correlate with known full-length FN proteins, the remainder of unconfirmed short or truncated FN transcripts may represent misaligned reads that originally derived from full-length transcripts. Future work will focus on additional analysis of mRNA and protein expression of specific alternatively spliced variants of FN to validate the results found in this study.

## 4. Conclusions

Based on reprocessed RNA-Seq data from the TOIL Recompute project, which sourced data from the TCGA and GTEx projects, we demonstrated that FN and its alternatively spliced isoforms exhibit high levels of overexpression in many different sites of primary tumors relative to corresponding normal tissues. In addition, we demonstrated correlations between FN expression and several clinical diagnostic and prognostic categories, including early-stage disease, lymph node activity, histological grade, and patient survival. Since oncofetal FNs are known to exhibit little to no protein expression in healthy normal tissues, the results from this study suggest that oncofetal FN presents a tumor-selective biomarker that can be exploited for targeted diagnostic and therapeutic agents for a multitude of cancers. Furthermore, diagnostic agents targeted to oncofetal FN have the potential to provide valuable diagnostic and prognostic information via non-invasive means, making them potentially high-value supplements to current diagnostic practices.

## 5. Methods

### 5.1. Data Sources

Bulk data, including RNA-Seq, protein expression, and clinical information, were obtained from the Toil Recompute project, available through the UCSC XENA web portal (https://xena.ucsc.edu/, accessed on 23 May 2020). The Toil Recompute project sourced their data from The Cancer Genome Atlas (TCGA) and the Genotype-Tissue Expression project (GTEx) [33,34].

### 5.2. Data Extraction and Processing

All data extraction and organization were completed in MatLab (MathWorks, Natick, MA, USA) by first loading the bulk datasets and matching sample IDs between RNA-Seq expression, protein expression, and clinical information. Datasets were then organized first by cancer type and then by tissue type (i.e., normal tissue, primary tumor, etc.). Cohort groups consisting only of primary tumor samples were then generated by separating the primary tumor datasets into various groups according to clinical information, such as pathologic N stage (i.e., N0 or N+) or histological grade (i.e., grades 1–4). Data were then exported in comma-separated variable (.csv) format for further analysis and the generation of figures.

RNA-Seq data from the Toil Recompute project are provided in the form of:$$Value=\log_{2}\left(TPM+0.001\right)$$

To recover the raw *TPM* expression value for each sample, the data values provided by the Toil Recompute project were processed as follows:$$TPM=2^{value}-2^{-9.9658}$$

Using raw *TPM* values allows for the direct summation of multiple transcript expression values within a sample. The sum of the *TPM* values of all 27 FN transcripts within a sample equals the *TPM* value for the *FN1* gene for that sample. Normalized *TPM* values were also produced when comparing the expression of various cohorts to a control reference across multiple cancer types. The normalization procedure was as follows:(1)$$Normalized\;TPM=\frac{TPM_{sample}}{TPM_{reference}}$$

Depending on the application, $TPM_{reference}$ refers to either the average expression in the respective normal tissue for the cancer type and the transcript group being analyzed, or the total expression of the *FN1* gene or ECM-FN in a sample. After processing, data were exported in comma-separated variable (.csv) format for further analysis and the generation of figures.

### 5.3. Statistical Analysis

Statistical analyses were performed in GraphPad Prism (GraphPad Software, San Diego, CA, USA). When analyzing averages of two groups across multiple cancer types, a paired *t*-test was used. When analyzing samples of two groups within a cancer type, an unpaired Welch *t*-test was used. When analyzing averages of more than two groups across multiple cancer types, a repeated-measures one-way ANOVA was used with the Dunnett post hoc test. When analyzing samples of more than two groups within a cancer type, a Brown–Forsythe and Welch one-way ANOVA was used with the Games–Howell post hoc test. Survival curves were generated according to standard Kaplan–Meier survival analysis, followed by log-rank statistical testing to determine differences between survival curves.

## Figures and Tables

**Figure 1 cells-12-00685-f001:**
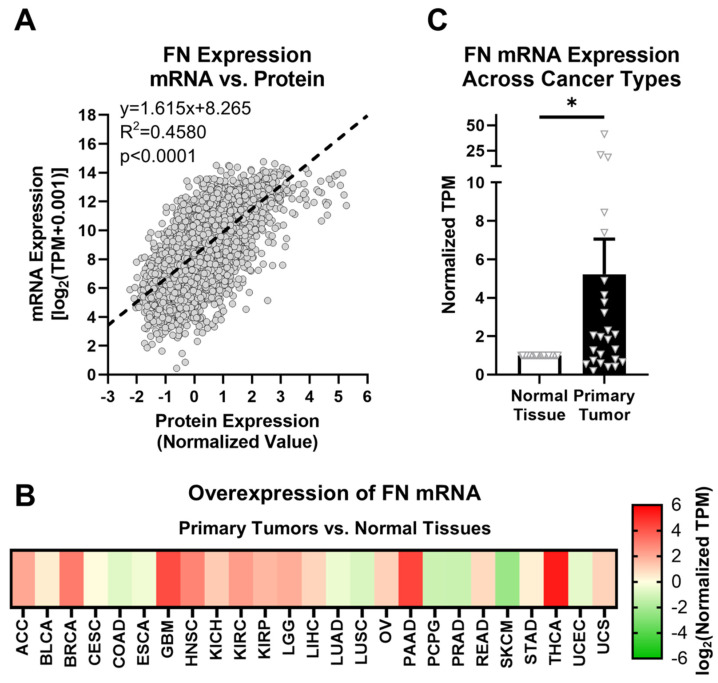
FN expression in primary tumors. (**A**) Correlation between FN protein and mRNA expression (*n* = 6539). (**B**) Overexpression of FN mRNA in primary tumors versus normal tissues (*n* = 25). Cancer types were excluded from analysis if the number of samples from either primary tumors or normal tissues was fewer than 10. (**C**) Average overexpression of FN mRNA in primary tissues versus normal tissues. Significance: *, *p* < 0.05.

**Figure 2 cells-12-00685-f002:**
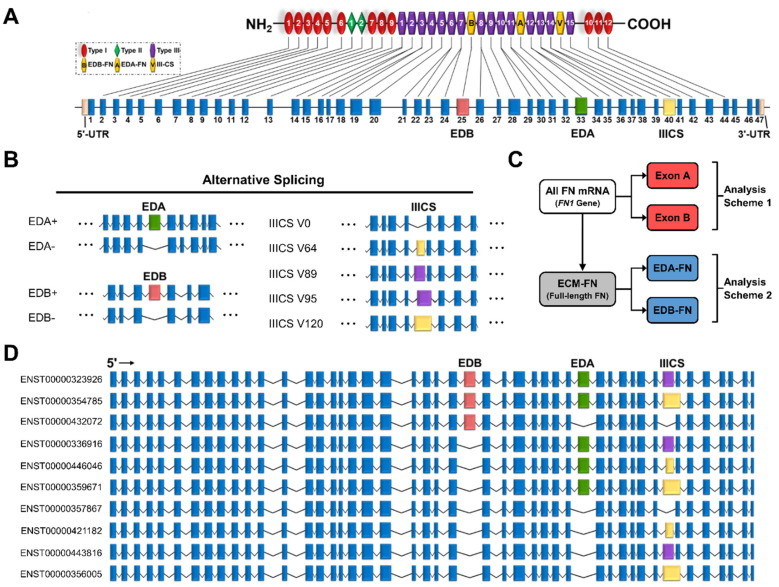
Transcriptome of the *FN1* gene. (**A**) FN protein and exon structure. The untranslated regions and exons that exhibit alternative splicing are in colors other than blue. (**B**) Alternative splicing of the EDA, EDB, and IIICS exons. The IIICS exon exhibits five splice variants of differing amino acid length indicated in the splice variant names. (**C**) Scheme displaying how transcripts are grouped for analysis. Exon A and Exon B are compared directly with all FN mRNAs (*FN1* genes), while EDA-FNs and EDB-FNs are compared directly with ECM-FNs. (**D**) Exon structures of the 10 full-length ECM-FNs recognized in the Ensembl library.

**Figure 3 cells-12-00685-f003:**
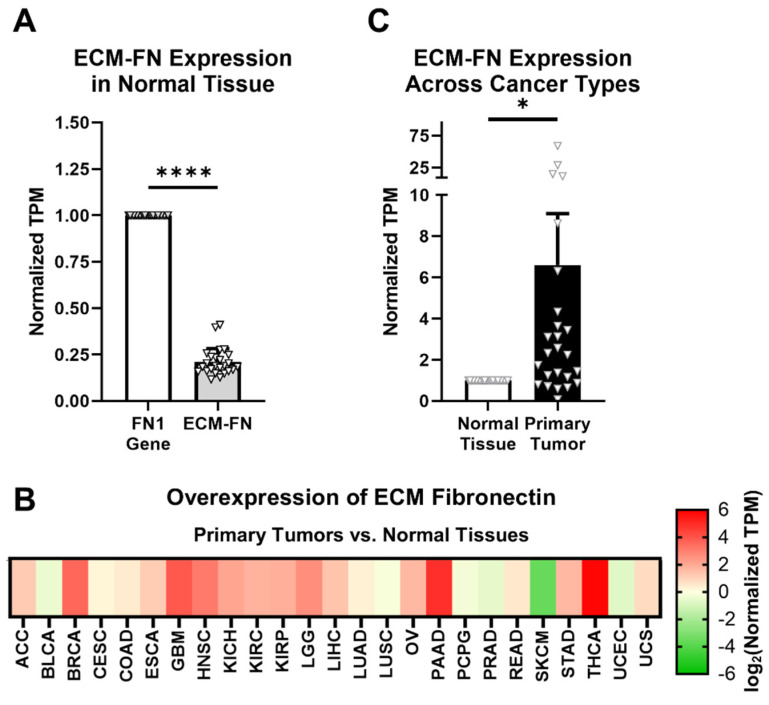
ECM-FN expression in primary tumors. (**A**) Expression of the 10 ECM-FN transcripts normalized to the whole *FN1* gene in primary tumors (*n* = 25). (**B**) Overexpression of ECM-FN mRNA in primary tumors versus normal tissues (*n* = 25). Cancer types were excluded from analysis if the number of samples in either the primary tumors or normal tissues was fewer than 10. (**C**) Average overexpression of ECM-FN mRNA in primary tissues versus normal tissues. Significance: *, *p* < 0.05; ****, *p* < 0.0001.

**Figure 4 cells-12-00685-f004:**
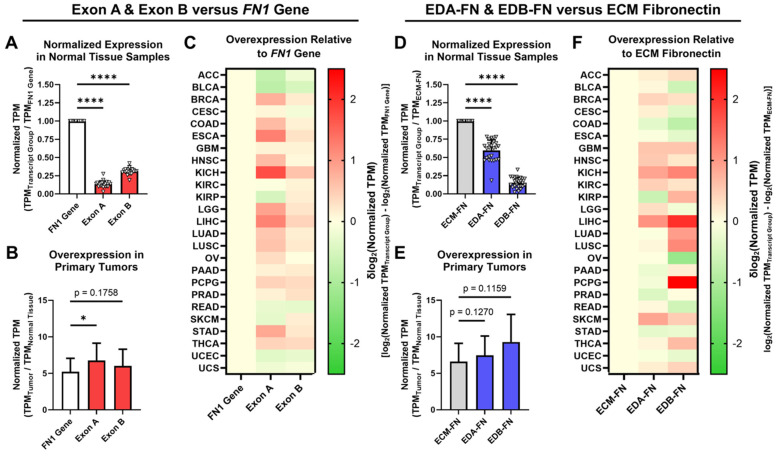
Expression of oncofetal FN in normal tissues and primary tumors. (**A**) Average expression of Exon A and Exon B normalized to the *FN1* gene in normal tissue samples (*n* = 25). (**B**) Average overexpression of the *FN1* gene, Exon A, and Exon B in primary tumors normalized to corresponding normal tissues (*n* = 25). (**C**) Heat map showing overexpression of Exon A and Exon B normalized to the *FN1* gene by cancer type. (**D**) Average expression of EDA-FN and EDB-FN normalized to ECM-FN in normal tissue samples (*n* = 25). (**E**) Average overexpression of ECM-FN, EDA-FN, and EDB-FN in primary tumors normalized to corresponding normal tissues (*n* = 25). (**F**) Heat map showing overexpression of EDA-FN and EDB-FN normalized to ECM-FN by cancer type. Tissue types were excluded from analysis if the number of samples in any cohort was fewer than 10. Significance: *, *p* < 0.05; ****, *p* < 0.0001.

**Figure 5 cells-12-00685-f005:**
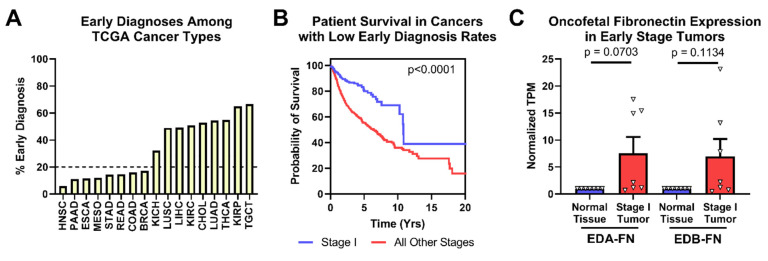
Oncofetal FN expression and early-stage diagnosis. (**A**) Percentage of patients diagnosed at stage I by cancer type. (**B**) Pan-cancer Kaplan–Meier survival curves of stage I patients (*n* = 363) compared to those diagnosed at all other stages (*n* = 2273) for cancer types with less than 20% of early-stage diagnosis. (**C**) Oncofetal FN expression in stage I tumors compared to corresponding normal tissue expression (*n* = 7). Cancer types were excluded from analysis if the number of normal tissue or stage I tumor samples was fewer than 10.

**Figure 6 cells-12-00685-f006:**
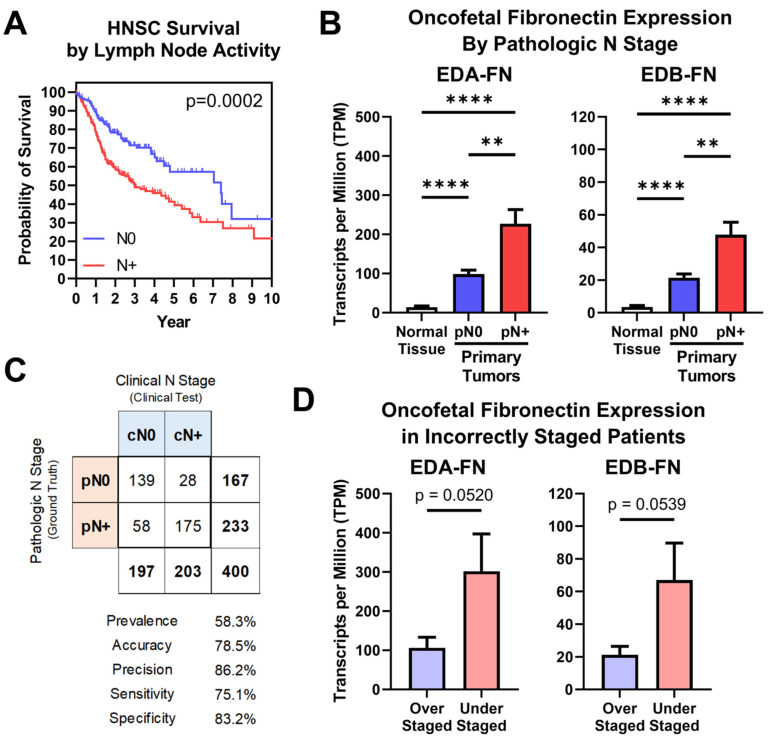
Oncofetal FN expression and lymph node activity. (**A**) Kaplan–Meier survival curves for HNSC patients diagnosed N0 (*n* = 174) and N+ (*n* = 243). (**B**) Oncofetal FN expression in normal HNSC tissue (*n* = 99) and primary tumors pathologically staged N0 (*n* = 174) or N+ (*n* = 243). (**C**) Binary classification comparing clinical N stage (cN; clinical test) with pathologic N stage (pN; ground truth). Binary classification statistics are provided. (**D**) Oncofetal FN expression in over-staged (*n* = 28) and under-staged (*n* = 58) HNSC patients. Significance: **, *p* < 0.01; ****, *p* < 0.0001.

**Figure 7 cells-12-00685-f007:**
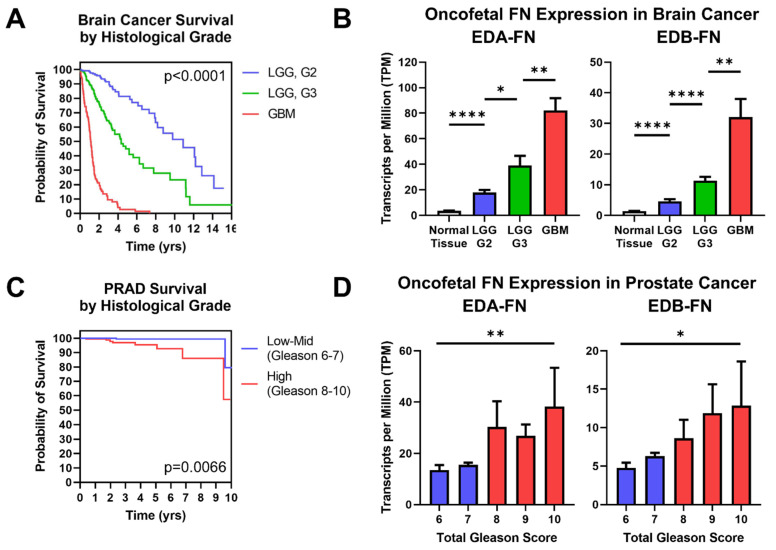
Oncofetal FN expression and histological grade. (**A**) Kaplan–Meier survival curves for brain cancers separated out by histological grade (LGG G2, *n* = 247; LGG G3, *n* = 260; GBM, *n* = 152). The colors of the curves correspond to the column graphs in (**B**). (**B**) Oncofetal fibronectin expression in brain cancers and corresponding normal tissue (*n* = 1152), separated out by histological grade (LGG G2, *n* = 247; LGG G3, *n* = 260; GBM, *n* = 152). Glioblastoma multiforme was considered the highest grade of brain cancer. (**C**) Kaplan–Meier survival curves for prostate cancer of low–mid grade (Gleason score 6–7; *n* = 271) and high grade (Gleason score 8–10; *n* = 191). Groups were combined for sample size purposes. The colors of the curves correspond to the column graphs in (**D**). (**D**) EDB-FN expression in prostate cancer separated out by total Gleason score (6, *n* = 43; 7, *n* = 228; 8, *n* = 57; 9, *n* = 131; 10, *n* = 4). Statistical analysis for prostate cancer data used the Kruskal–Wallis test. Significance: *, *p* < 0.05; **, *p* < 0.01; ****, *p* < 0.0001.

**Figure 8 cells-12-00685-f008:**
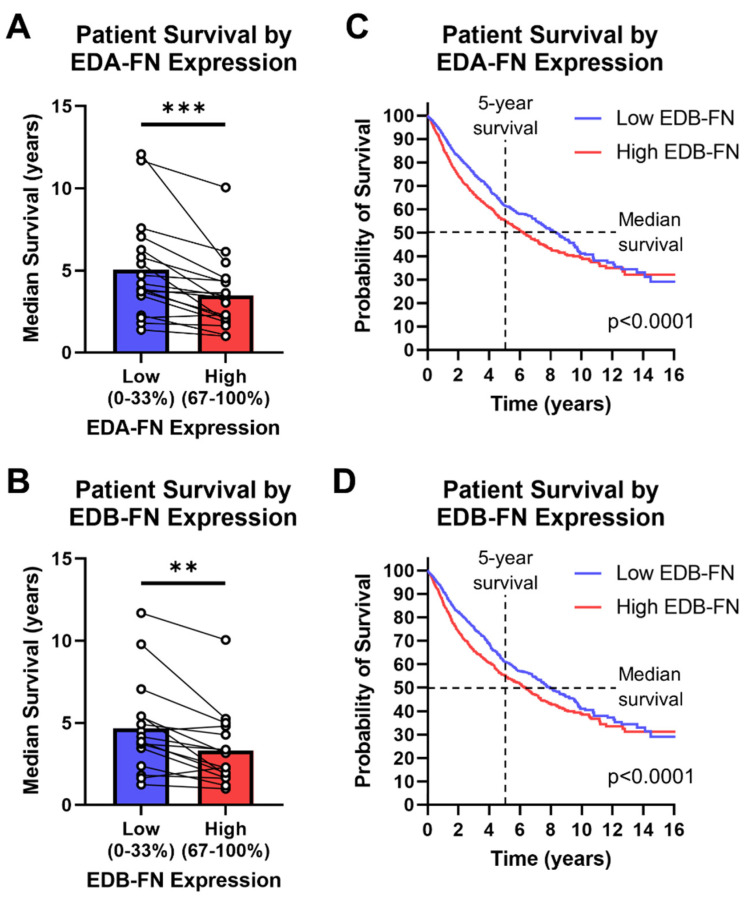
Oncofetal FN expression and patient prognosis. Median patient survival time based on high (top 33%) or low (bottom 33%) (**A**) EDA-FN (*n* = 18) and (**B**) EDB-FN (*n* = 16) expression within each cancer type. Pan-cancer Kaplan–Meier survival curves comparing patients expressing low and high levels of (**C**) EDA-FN (low, *n* = 3055; high, *n* = 3055) and (**D**) EDB-FN (low, *n* = 3051; high, *n* = 3046). Low- and high-expression groups were determined within each cancer type before being combined into a single dataset. Significance: **, *p* < 0.01; ***, *p* < 0.001.

**Table 1 cells-12-00685-t001:** **TCGA study names.** Table of the diseases studied in the TCGA project, including disease abbreviations and full disease names. Tissues studied in the GTEx project that directly correlate with TCGA disease locations are also shown.

TCGA Cancer Types		
ABBV	Full Disease Name	GTEx Tissues
ACC	Adrenocortical carcinoma	Adrenal gland
BLCA	Bladder urothelial carcinoma	Bladder
BRCA	Breast invasive carcinoma	Breast
CESC	Cervical squamous cell carcinoma and endocervical carcinoma	Cervix uteri
CHOL	Cholangiocarcinoma	
COAD	Colon adenocarcinoma	Colon
DLBC	Diffuse large B-cell lymphoma	
ESCA	Esophageal carcinoma	Esophagus
GBM	Glioblastoma multiforme	Brain
HNSC	Head and neck squamous cell carcinoma	Salivary gland
KICH	Kidney chromophobe	Kidney
KIRC	Kidney renal clear cell carcinoma	Kidney
KIRP	Kidney renal papillary cell carcinoma	Kidney
LAML	Acute myeloid leukemia	Bone marrow
LGG	Brain lower glade glioma	Brain
LIHC	Liver hepatocellular carcinoma	Liver
LUAD	Lung adenocarcinoma	Lung
LUSC	Lung squamous cell carcinoma	Lung
MESO	Mesothelioma	
OV	Ovarian serous cystadenocarcinoma	Ovary
PAAD	Pancreatic adenocarcinoma	Pancreas
PCPG	Pheochromocytoma and paraganglioma	Nerve
PRAD	Prostate adenocarcinoma	Prostate
READ	Rectum adenocarcinoma	
SARC	Sarcoma	
SKCM	Skin cutaneous melanoma	Skin
STAD	Stomach adenocarcinoma	Stomach
TGCT	Testicular germ cell tumors	
THCA	Thyroid carcinoma	
THYM	Thymoma	
UCEC	Uterine corpus endometrial carcinoma	Uterus
UCS	Uterine carcinosarcoma	Uterus
UVM	Uveal melanoma	

## Data Availability

The raw data is available at https://xena.ucsc.edu, accessed on 8 February 2023.

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
