# Peer review of "RNA-Seq Analysis of Extradomain A and Extradomain B Fibronectin as Extracellular Matrix Markers for Cancer"

_cells, 2023, doi:10.3390/cells12050685_

Round 1
Reviewer 1 Report
Revision for “RNA-seq analysis of extradomain A and extradomain B fibronectin as extra cellular matrix markers for cancer”
In this manuscript Hall and colleagues want to demonstrate fibronectin and especially oncofetal fibronectins are valid biomarkers for several different tumors as their expression correlates with tumor stage, lymph node activity, histological grade and patient survival. The entire study relies on the analysis of TCGA and GTEx data (RNA-seq, protein expression and clinical information) through MatLab and GraphPad Prism software.
The subject is interesting but there is a lot of previous literature proving the same findings with alternative methodologies (https://doi.org/10.1002/1878-0261.12705, https://doi.org/10.1016/j.addr.2015.11.014, http://dx.doi.org/10.1136/jitc-2021-004479, https://doi.org/10.7150/thno.44948). The only novelty part lays in the overall analysis of the aforementioned parameters in 32 different tumors although the correlation to lymph node activity and histological grade were assessed only in HNSC and LGG/GBM, respectively. Diversification between all transcripts of the fibronectin gene and only the full-length isoforms was also newly addressed.
Some of the data, for example the one presented in Fig 6, are not clear and need to be better discussed in both results and discussion sections.
Although the aim of the work is to prove fibronectin as a universal cancer biomarker, the data seem to suggest that especially ECM-FN, EDA and EDB transcripts could improve the early diagnosis of specific types of tumors. Focusing on them would improve the strength of many statistical analyses presented.
Regarding the specific quality of work those are my comments:
Major comments:
· Number of patients analyzed is a crucial factor and need to be disclosed for each analysis and not just for some as the authors are doing in the submitted version of this manuscript
Minor comments:
· The correlation founds in Figure 1A is very strong and convincing, and it is found comparing the mRNA and Protein expression in all the tumors analyzed. When authors analyze a specific type of tumor, like in Figure 6, they need to prove first that also in this peculiar disease the correlation found in Fig 1 A is valid, performing the same kind of analysis.
· Fig. 4B and Fig. 4E: it is not clear whether the data presented are relative to the overexpression of the transcripts over the normalized expression in normal tissue samples. Pleas explain and fix figure and legend
· Legend Fig. 4A (rows 185-186) misses “in normal tissue samples”
· Legend Fig. 4B (rows 186-187) misses “in primary tumor samples”
· Legend Fig. 4D (rows 188-189) misses “in normal tissue samples”
· legend Fig. 4E (row 189) misses “in primary tumor samples”
· Row 218: typo
Author Response
Summary: In this manuscript Hall and colleagues want to demonstrate fibronectin and especially oncofetal fibronectins are valid biomarkers for several different tumors as their expression correlates with tumor stage, lymph node activity, histological grade and patient survival. The entire study relies on the analysis of TCGA and GTEx data (RNA-seq, protein expression and clinical information) through MatLab and GraphPad Prism software. The subject is interesting but there is a lot of previous literature proving the same findings with alternative methodologies (https://doi.org/10.1002/1878-0261.12705, https://doi.org/10.1016 /j.addr.2015.11.014, http://dx.doi.org/10.1136/jitc-2021-004479, https://doi.org/10.7150/thno.44948). The only novelty part lays in the overall analysis of the aforementioned parameters in 32 different tumors although the correlation to lymph node activity and histological grade were assessed only in HNSC and LGG/GBM, respectively. Diversification between all transcripts of the fibronectin gene and only the full-length isoforms was also newly addressed. Some of the data, for example the one presented in Fig 6, are not
clear and need to be better discussed in both results and discussion sections. Although the aim of the work is to prove fibronectin as a universal cancer biomarker, the data seem to suggest that especially ECM-FN, EDA and EDB transcripts could improve the early diagnosis of specific types of tumors. Focusing on them would improve the strength of many statistical analyses presented.
Thank you for the review and comments. Regarding the data presented in Fig. 6, we have added a supplemental figure (Supp. Fig. 3) detailing the structure of the confusion matrix and calculation of the provided statistics. The structure of the results section has additionally been altered to improve clarity, and the discussion section regarding this topic has been expanded.
Comments: Regarding the specific quality of work those are my comments:
Major comments:
- Number of patients analyzed is a crucial factor and need to be disclosed for each analysis and not just for some as the authors are doing in the submitted version of this manuscript.
Patient cohort sizes are indicated in Supplementary Table 1 in the supplemental information. We will keep this table in supplementary information, and have updated figure legends to indicate sample sizes where the supplementary table does not indicate such. This includes sample sizes when focusing on averages across cancer types (i.e. Fig. 1B, 2A, 2C, etc.), averages among patients when not taking cancer type into account (i.e. Fig. 1A), and survival curves (i.e. Fig. 5B, 6A, etc.).
Minor comments:
- The correlation founds in Figure 1A is very strong and convincing, and it is found comparing the mRNA and Protein expression in all the tumors analyzed. When authors analyze a specific type of tumor, like in Figure 6, they need to prove first that also in this peculiar disease the correlation found in Fig 1 A is valid, performing the same kind of analysis.
The analysis in Figure 1 followed a pan-cancer method to establish an overall positive trend between fibronectin mRNA and protein expression. This was done, in part, because several cancer types in the dataset have a significantly smaller sample size than others. The mRNA-protein trends have been separated out by cancer type and included as Supplementary Figure 1, and the text has been updated accordingly.
- 4B and Fig. 4E: it is not clear whether the data presented are relative to the overexpression of the transcripts over the normalized expression in normal tissue samples. Please explain and fix figure and legend. Legend Fig. 4A (rows 185-186) misses “in normal tissue samples”. Legend Fig. 4B (rows 186-187) misses “in primary tumor samples”. Legend Fig. 4D (rows 188-189) misses “in normal tissue samples”. Legend Fig. 4E (row 189) misses “in primary tumor samples”.
Thank you for pointing out the confusion surrounding this figure. Figures 4B and 4E are showing average overexpression of general FN and oncofetal variants in primary tumors relative to corresponding normal tissue samples. Figures 4C and 4F then look at these on a cancer-by-cancer basis, normalizing the overexpression values to those of general FN to demonstrate the higher overexpression of the oncofetal variants. The figure legend and y-axis labels have been updated to improve figure clarity.
- Row 218: typo
The referenced typo (“48%.38”) originates from the production of the PDF file by Cells. The 48% references a clinical statistic, which was pulled from citation #38.
Reviewer 2 Report
In this interesting manuscript authors described the potential role of oncofetal fibronectins as cancer biomarkers. Using RNA-seq data from the Cancer Genome Atlas Program and the Genotype-tissue Expression Project databases they found a significant over-expression of alternatively spliced fibronectin isoforms in several tumors, which well-correlated with stage, lymph node activity and histological grade at diagnosis. The paper is well written and clear. The main weakness is that conclusions in this work were drawn from reanalysis data not followed by experimental confirmation. In addition, the role of oncofetal Fibronectins as tumor-biomarkers is not novel. My major concern is that major findings should be verified by biochemical/histological approaches. Could it be possible to test specific EDA/EDB antibodies on tissue specimens? In general, clinical correlation with EDA or EDB expression are comparable. However, EDA and EDB-containing FNs should impact the tumor pathobiology in a different way. How do the authors explain similar results? Sample numbers included in all analyses should be indicated.
Author Response
Summary: In this interesting manuscript authors described the potential role of oncofetal fibronectins as cancer biomarkers. Using RNA-seq data from the Cancer Genome Atlas Program and the Genotypetissue Expression Project databases they found a significant overexpression of alternatively spliced fibronectin isoforms in several tumors, which well-correlated with stage, lymph node activity and histological grade at diagnosis. The paper is well written and clear. The main weakness is that conclusions in this work were drawn from reanalysis data not followed by experimental confirmation. In addition, the role of oncofetal Fibronectins as tumor-biomarkers is not novel.
We appreciate the comments. The goal of this work is to perform bioinformatic analysis of the markers in all available data. We are aware of the limitation work. Nevertheless, the analysis will provide valuable information for future comprehensive experimental validation of the markers.
Comments:
- My major concern is that major findings should be verified by biochemical/histological approaches. Could it be possible to test specific EDA/EDB antibodies on tissue specimens?
Thank you for your concern regarding this topic – we also identified this as a significant limitation of the study in the discussion. Due to the scale of the additional experiments required to validate the results (i.e. via biochemical or histological approaches), we noted that the results presented in this manuscript will require future validation in follow-up publication(s). Validation of all results presented in this manuscript would likely take in excess of a year, possibly multiple years, due to tissue procurement and relevant protein-based assays for all cancer types and cohorts analyzed via the RNA-Seq methodology. This is unfortunately not feasible within the timeframe for resubmission and will be the focus of follow-up publication(s).
- In general, clinical correlation with EDA or EDB expression are comparable. However, EDA and EDB-containing FNs should impact the tumor pathobiology in a different way. How do the authors explain similar results?
In general, we have found that EDA-FN and EDB-FN mRNA levels increase as general FN increases, although to varying degrees. We agree that EDA-FN and EDB-FN may have unique and necessary functions in cancer. The functions of both EDA and EDB are still being studied to better elucidate their roles in cancer biology. In addition, cancer is a highly heterogeneous disease, with the possibility for EDA-FN, EDB-FN, or both to exhibit substantial upregulation depending on the tumor microenvironment and the particular roles of the oncofetal FN variants.
- Sample numbers included in all analyses should be indicated.
Patient cohort sizes are indicated in Supplementary Table 1 in the supplemental information. We will keep this table in supplementary information, and have updated figure legends to indicate sample sizes where the supplementary table does not indicate such. This includes sample sizes when focusing on averages across cancer types (i.e. Fig. 1B, 2A, 2C, etc.), averages among patients when not taking cancer type into account (i.e. Fig. 1A), and survival curves (i.e. Fig. 5B, 6A, etc.).
Round 2
Reviewer 2 Report
All my comments have been addressed. I have no further remarks